# Evaluating the technical efficiency and influencing factors of citrus fruits planting in China

Yuan Wu *, Xiaojun Pu

Institute of Agricultural Science and Technology Information, Chongqing Academy of Agricultural Sciences, Chongqing, China

* 372495245@qq.com

## Abstract

By adopting the stochastic frontier analysis model of a beyond logarithmic production function and utilizing panel data for citrus planting operations in seven major citrus-producing provinces and cities in China from 2012 to 2024, the technical efficiency of citrus fruit planting in China was measured, and the influencing factors were analyzed. The results show that:(1) The technical efficiency of citrus planting in the main producing provinces and cities shows an overall trend of growth followed by a slow decline, then slow growth followed by decline, and finally tends to be stable, with an overall average technical efficiency of 0.837. (2) Regional disparities exist in the technical efficiency of citrus planting. The technical efficiency of citrus planting in the eastern region surpasses that of other regions significantly. Since 2018, the average technical efficiency in central and western regions has shown convergent trends. (3) The increase of labor inputs, direct material and service inputs, mainly fertilizers and pesticides, and indirect material and service inputs, mainly depreciation of fixed assets and sales fees, has a positive effect on the output, but under the premise of unchanged labor and capital inputs, citrus outputs will decline with the time. (4) There is an economy of scale effect, but if the scale is too large, the technical efficiency of planting will be reduced instead. Technological progress plays a crucial role in citrus planting, relying solely on the growth of inputs, such as chemical fertilizers, agriculture, and labor, will lead to a loss of efficiency. Increasing the scale of production will help to improve the technical efficiency of citrus production. Citrus planting needs to convert new growth momentum, the key to which is to increase investment in agricultural science and technology innovation, promoting industrial innovation through scientific and technological innovation. It is necessary to improve the monitoring for, and provide of early warning of, natural disasters in agriculture, increase the standardization of citrus orchards, and continuously improve the efficiency of citrus orchard planting and management.

**Data availability statement:** All relevant data are within the paper and its Supporting information files.

**Funding:** This research was funded by the Open Fund of Key Laboratory of Agricultural Monitoring and Early Warning Technology, Ministry of Agriculture and Rural Affairs (No. 2023KLAMEWT01).

**Competing interests:** NO authors have competing interests Enter: The authors have declared that no competing interests exist.

## Introduction

Citrus, indigenous to China, stands as one of the world's premier cash crops, renowned for its high economic significance and nutritional richness. It has been cultivated for more than 4,000 years, and as the level of citrus planting technology has improved, the production of citrus has been increasing. Global citrus production reached 198 million tons in 2024 [1], representing an increase of nearly 30% over the past decade. Citrus is now the world's number one fruit and the third most traded agricultural commodity, after wheat and maize. Citrus is the number one fruit in China, ranking first in the world in terms of planted area and output year-round, and is the fruit tree with the widest cultivated area and the most important economic status in the south of China; it has become the most important sector in China's southern economy of fruits and forests, and it has a very important position in the revitalization of the countryside [2]. In 2024, China's citrus planting area reached 46.54 million mu, and the output reached 67.92 million tonnes [3]. While the citrus industry is booming, citrus planting is generally facing problems such as climate impact, disease and pest control, and low management efficiency. Although China's citrus planting area and total output ranked first in the world, but the trade volume accounted for a small proportion of the world. While China holds the title of the world's largest citrus producer and trader, it does not yet qualify as a powerful or dominant producer and trader in the field. In comparison to developed countries that specialize in citrus production and major citrus-exporting nations, the production efficiency of Chinese citrus remains relatively low. The average yield of Chinese citrus reached 21887.6 kg/ha in 2024, which is equal to the world's average yield level, but compared with Israel, the average yield level of China is only 50% of theirs. The main reason for this is related to the form of operation and the efficiency of the production and operation of citrus planting. Therefore, the technical efficiency of citrus planting in China and the factors affecting it are issues that deserve attention and research. Measuring the technical efficiency of citrus planting in China and analyzing the main factors affecting the technical efficiency of citrus planting are important references for improving the efficiency of citrus planting in China, transforming the mode of operation, and enhancing the competitiveness of the industry.

Efficiency in economics typically refers to the state of achieving the maximum output with a given set of inputs or the minimum inputs to achieve a specified output. When evaluating agricultural production efficiency, the primary methods can be categorized into two main groups: non-parametric and parametric. Specifically, non-parametric methods primarily rely on the Data Envelopment Analysis (DEA) model [4], while parametric methods mainly utilize the Stochastic Frontier Analysis (SFA) method [5]. Furthermore, in the field of agricultural production efficiency research, numerous studies have employed DEA and its extended models, the Malmquist index method, and other techniques for in-depth analysis.

A study on the production efficiency of organic agriculture in Spain found that the low efficiency of this production model is mainly due to insufficient output [6]. An analysis of the efficiency of organic citrus orchards using the DEA model revealed that policy regulations, planting technology, the level of education of producers, and

agricultural experience have a significant direct impact on technical efficiency [7,8]. In addition, a DEA assessment of environmental issues in citrus planting on farms indicated that excessive use of nitrogen fertilizers is a major cause of inefficiency [9]. Furthermore, a study based on 1,009 observation samples from 11 countries specifically analyzed the efficiency of irrigation water and nitrogen fertilizer use in citrus planting. The study found that optimizing irrigation water management and rational application of nitrogen fertilizer can significantly increase citrus yields [10]. A study using both DEA and SFA models to measure technical efficiency and scale efficiency in Italy's citrus planting industry found that the technical efficiency estimated by the SFA model was roughly on par with that estimated by the DEA model, while the scale efficiency estimated by the SFA model was higher than that derived from the DEA model [11].

In terms of research on the planting efficiency of citrus in China, the use of the Malmquist index method to measure changes in total factor productivity (TFP) indicates that most provinces and regions exhibit a coexistence of techno-logical progress and agricultural efficiency losses [12]. By employing the DEA method to calculate planting efficiency by variety and region, the results reveal that excessive investment in pesticide costs and other material expenses is the key factor contributing to the relatively low production efficiency of citrus in China [13]. The application of the DEA-Malmquist index method to measure the TFP of China's citrus indicates that technological progress is the dom-inant factor influencing citrus TFP, while pure technical efficiency and scale efficiency are important factors affecting TFP [14]. A study employing the Malmquist index method to calculate and analyze China's citrus TFP, combined with the Tobit model, analyzed the external factors influencing the growth of citrus TFP [15]. At the regional level, a study focusing on the citrus industry in Hubei Province found through the DEA method that changes in the scale efficiency of citrus production in the province determine changes in comprehensive technical efficiency [16]; A study using the citrus industry in Chongqing as an example employed the DEA method to analyze planting efficiency across multiple scale intervals in citrus cultivation, finding that there are multiple efficient decision intervals where input-output ratios are in equilibrium [17].

From the existing literature, it is evident that both approaches to efficiency research have been considered in the study of citrus production efficiency problems, with domestic studies primarily focusing on the use of non-parametric methods, predominantly employing DEA models. However, this type of methodology has its own technical shortcomings. Although Chen Xinjian et al. used the stochastic frontier analysis method, the empirical analysis sample only considered the his-torical data of five provinces and failed to consider the western citrus planting areas (Guangxi, Chongqing, etc.), and less consideration was given to the environment of citrus planting in analyzing the efficiency influencing factors [18].Agricultural production is highly influenced by stochastic factors, including natural geography, climate, and market fluctuations, making the use of stochastic frontier analysis more appropriate than traditional DEA. From the research perspective, there is more literature on the trend of citrus production efficiency changes in a certain period, and less literature on the in-depth anal-ysis of efficiency influencing factors. In view of this, this study adopts the stochastic frontier analysis method, using the latest statistical data, to measure the technical efficiency of citrus planting in seven major regions of China (Fujian, Guang-dong, Hubei, Hunan, Jiangxi, Chongqing, and Guangxi), and innovatively incorporates the natural and socio-economic development conditions such as climatic conditions, regional economic disparities, and transportation infrastructures into the non-efficiency influencing factors for empirical analysis, in order to provide theoretical references for the enhancement of citrus planting efficiency and industrial competitiveness.

## Materials and methods

### Model construction

According to Aigner [19], Battese [20], and others [21–23], the stochastic frontier production function based on panel data is generally expressed by the following equation:

$$Y_{it} = \beta X_{it} + (V_{it} - U_{it}) \tag{1}$$

where $Y_{it}$ represents the output vector of sample $i$ in period $t$. $X_{it}$ represents the vector of inputs of sample $i$ in period $t$. $\beta$ is the parameter to be estimated, $V_{it}$ stands for the random error term, which represents the uncontrollable factor that follows a normal distribution with mean zero and variance $\sigma_\nu$, $U_{it}$ is a non-negative random variable representing the efficiency loss of the production system and is used to calculate technical inefficiency, which is assumed to follow a semi-normal, lognormal, or truncated normal distribution, $V_{it}$ and $U_{it}$ are independent of each other [24].

Citrus planting is a typical labor-intensive task [25], and from a practical point of view, citrus harvests are affected by labor intensity, fertilizer and pesticide inputs, etc., based on the current situation and the caliber of agricultural statistics and data availability. According to the current situation and the caliber of agricultural statistics and the availability of data, this paper chooses the output value of citrus ($Q_{it}$) to portray the output indicators of the production system, and selects labor input ($L_{it}$), direct material and service input ($K_{it}$), indirect material and service input ($S_{it}$) as the system. The direct material and service inputs include pesticide, fertilizer, seedlings, drainage and irrigation. while the indirect material and service inputs include insurance, management, and sales inputs. The stochastic frontier production function can be expressed as:

$$Q_{it} = f(L_{it}, K_{it}, S_{it}; T) + V_{it} - U_{it} \tag{2}$$

In Eq. (2), $T$ represents the time factor, $U_{it}$ represents the technical inefficiency term, and $V_{it}$ represents the randomly distributed term. When there is no technical inefficiency term, i.e., when $U_{it}=0$, it means that the production system has reached the optimal frontier level [26]. Therefore, the technical efficiency function $TE_{it}$ can be constructed and its expression is as follows:

$$TE_{it} = \frac{E\left[f\left(L_{it}, K_{it}, S_{it}, T; \beta\right)\right] \exp(V_{it} - U_{it})}{E\left[f\left(L_{it}, K_{it}, S_{it}, T; \beta\right)\right] \exp\left(V_{it}\right)} = \exp(-U_{it}) \tag{3}$$

The stochastic frontier analysis method is differs from DEA, which requires implementation to determine the form of the production function. In the specific production function selection, due to the Cobb-Douglas production function premise assumptions being strong, it cannot effectively distinguish between random noise and technological advances. A large number of studies have shown that the transcendental logarithmic function is more inclusive, the form is flexible, and can be better fitted to the data. In particular, it is able to effectively deal with the non-equilibrium or heterogeneous types of data, and it can reflect the explanatory variables on the interaction of the explanatory variables. Therefore, this paper chooses to construct the analytical model with the beyond logarithmic production function, and takes the logarithm of both sides of Equation (2) to expand as follows:

$$\begin{aligned}
\ln Q_{it} = {} & \beta_0 + \beta_1 t + \tfrac{1}{2}\beta_2 t^2 + \beta_3 \ln L_{it} + \beta_4 \ln K_{it} + \beta_5 \ln S_{it} + \beta_6 t \times \ln L_{it} + \beta_7 t \times \ln K_{it} + \\
& \beta_8 t \times \ln S_{it} + \tfrac{1}{2}\beta_9 \ln L_{it} \times \ln K_{it} + \tfrac{1}{2}\beta_{10} \ln L_{it} \times \ln S_{it} + \tfrac{1}{2}\beta_{11} \ln K_{it} \times \ln S_{it} + \\
& \tfrac{1}{2}\beta_{12}(\ln L_{it})^2 + \tfrac{1}{2}\beta_{13}(\ln K_{it})^2 + \tfrac{1}{2}\beta_{14}(\ln S_{it})^2 + V_{it} - U_{it}
\end{aligned} \tag{4}$$

In order to test the reasonableness of the model, a statistic $\gamma = \frac{\sigma_U^2}{\sigma_U^2 + \sigma_V^2} (0 \leq \gamma \leq 1)$ was constructed. The larger the value of $\gamma$, the higher the degree to which the model is able to explain the loss of technical efficiency in citrus production and the more reasonable the model is.

The production activities of citrus planting are profoundly affected by the impacts of the natural and social environments. For example, climate has a significant impact on citrus planting [27]. Citrus frost damage not only affects plant production but also has a sustained impact on production over a number of years, and the problem of transport accessibility in citrus-producing areas has a direct impact on whether citrus crops can be marketed successfully [28]. In order to objectively describe the factors that may affect the inefficiency of citrus planting, based on the actual situation of production,

combined with previous relevant studies, and the availability of data, the regional economic differences, climatic differences, and transportation infrastructure conditions are included in the analytical framework, and the inefficiency function is constructed as follows:

$$U_{it} = \beta_{15}A_{it} + \beta_{16}I_{it} + \beta_{17}W_{it} + + \beta_{18}F_{it} + \beta_{19}H_{it} + \beta_{20}D_{it} \tag{5}$$

where a represents the regional variable, so that East = 1, Central = 2, West = 3; $I_{it}$ represents the per capita GDP of citrus production areas, which is used to portray the level of economic development and consumption capacity of citrus production areas; $W_{it}$ represents the level of employment, the cost of hiring labor compared to the total cost of labor, which is used to portray the scale of citrus production. Citrus production is a labor-intensive activity, and citrus harvesting, fertilizer application, pesticide spraying, and other processes are highly dependent on the labor force. The higher the level of employment, the larger the scale of cultivation, the lower the level of employment, the smaller the scale of cultivation. $F_{it}$ represents the financial support to agriculture, and the ratio of agriculture, forestry, and water expenditure to the local financial expenditure, reflecting the local government's financial support to citrus production. $H_{it}$ represents the density of the road network, which is used to reflect the status of the road infrastructure, and $D_{it}$ represents the incidence rate of meteorological disasters, reflecting the natural influences affecting the production of citrus fruit.

## Data sources

Citrus planting in China has obvious regional characteristics. It currently has several distinct citrus belts: the upper and middle reaches of the Yangtze River citrus belt, the Gannan-Xiangnan citrus belt, the Zhejiang, Fujian, and Guangdong citrus belt, the Exi-Xiangxi citrus belt, and the Xijiang late-maturing citrus belt. Citrus fruits are mainly planted in areas such as Fujian, Guangdong, Hubei, Hunan, Jiangxi, Chongqing, and Guangxi, as well as other provinces and municipalities (Fig 1). To ensure data continuity and availability, the above seven provinces and municipalities were selected as sample regions for this study. Their planting operation-related statistics over the past 13 years were collected and collated to assemble panel data. From the provincial perspective, the area and production of citrus cultivation in these seven regions account for more than 90 percent of national production. Therefore, the selected data are representative.

The data on citrus output value ($Q_{it}$), labor input ($L_{it}$), direct material and service input ($K_{it}$), indirect material and service input ($S_{it}$), and the level of hired labor ($W_{it}$) were obtained from the National Compendium of Agricultural Product Costs and Benefits (2013–2025), while the data on per capita regional GDP ($I_{it}$) and road network density ($H_{it}$) were obtained from the online database of the National Bureau of Statistics (http://data.stats.gov.cn/), financial support for agriculture ($F_{it}$) from China Statistical Yearbook (2013–2025), and meteorological disaster incidence ($D_{it}$) from China Rural Statistical Yearbook (2013–2025). In order to eliminate the effect of price changes, the consumer price index is used to calculate the value of the relevant value quantities in the sample. The data presented in Table 1 are descriptive statistics of citrus fruits.

As can be seen from Table 1, there are large regional differences in output value per mu, labor input, direct material and service input, indirect material and service input, GDP per capita, hiring rate, financial support for agriculture, road network density, and disaster rate in the seven main citrus fruit producing areas. Among these, the standard deviation of the output value per mu is the largest, which indicates that it is the most fluctuating; the second most fluctuating is the direct material and service input per mu, and the standard deviation of the road network density is the smallest, indicating that it is less volatile and tends to be stable. The standard deviation of indirect material and service inputs per mu exceeds the mean, indicating a high degree of data dispersion, the presence of outliers, and a skewed distribution.

## Model estimation and analysis

According to the set beyond the logarithmic production function and inefficiency function, the frontier4.1 software is used to estimate the relevant parameters, and the estimation results are shown in Table 2. As can be seen from Table 2, the $\sigma^2$

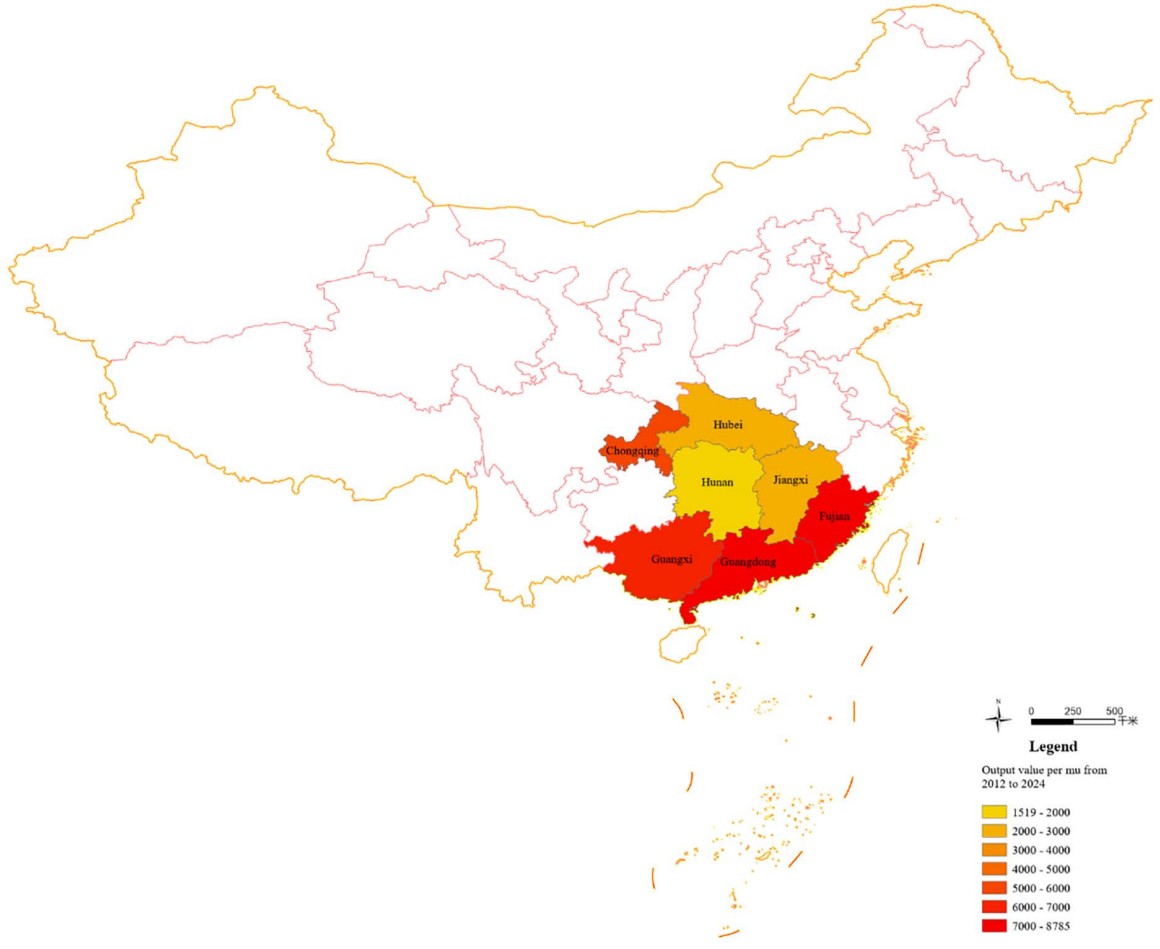

**Fig 1. Output value per mu in seven major citrus-producing regions from 2012 to 2024.**

**Table 1. Descriptive statistics of data for each indicator in the study sample.**

| Indicator name | Mean value | Standard deviation | Minimum value | Extreme value |
|---|---|---|---|---|
| Output value per mu (yuan) | 4494.627 | 2722.546 | 734.117 | 11306.672 |
| Labor input per mu (yuan) | 1567.258 | 911.227 | 634.670 | 4899.943 |
| Direct material and service input per mu (yuan) | 1121.089 | 931.614 | 196.475 | 3727.830 |
| Indirect material and service input per mu (yuan) | 240.447 | 247.596 | 39.090 | 894.860 |
| Gross regional product per capita (million yuan) | 5.531 | 2.129 | 2.380 | 11.270 |
| Employment rate (%) | 30.200 | 24.000 | 0.000 | 86.900 |
| Financial support for agriculture (%) | 10.000 | 2.300 | 5.000 | 14.600 |
| Road network density (km/sq km) | 1.181 | 0.426 | 0.456 | 2.271 |
| Disaster rate (%) | 3.600 | 3.100 | 0.400 | 14.700 |
| Region (East = 1, Centre = 2, West = 3) | 2.000 | 0.760 | 1.000 | 3.000 |

**Table 2. Stochastic frontier function estimation results.**

| Coefficient | Item | Estimated value | Standard deviation | T-value |
|---|---|---|---|---|
| $\beta_0$ (intercept) | | -44.536 | 95.380 | −0.467 |
| $\beta_1$ | $t$ | -0.015 | 0.172 | −0.881 |
| $\beta_2$ | $t^2$ | 0.003*** | 0.0002 | −14.664 |
| $\beta_3$ | $\ln L$ | -9.123 | 22.643 | −0.403 |
| $\beta_4$ | $\ln K$ | 46.332*** | 1.112 | 41.680 |
| $\beta_5$ | $\ln S$ | -7.929 | 10.880 | −0.729 |
| $\beta_6$ | $t\ln L$ | -0.335*** | 0.024 | −13.413 |
| $\beta_7$ | $t\ln K$ | 0.452*** | 0.013 | 33.929 |
| $\beta_8$ | $t\ln S$ | -0.008 | 0.007 | −1.047 |
| $\beta_9$ | $\ln L \cdot \ln K$ | -0.036 | 0.131 | −0.248 |
| $\beta_{10}$ | $\ln L \cdot \ln S$ | 0.231*** | 0.013 | 19.398 |
| $\beta_{11}$ | $\ln K \cdot \ln S$ | -0.898*** | 0.043 | −21.086 |
| $\beta_{12}$ | $(\ln L)^2$ | 0.327 | 1.767 | 0.185 |
| $\beta_{13}$ | $(\ln K)^2$ | -2.716*** | 0.062 | −44.051 |
| $\beta_{14}$ | $(\ln S)^2$ | -0.022 | 0.050 | −0.437 |
| $\beta_{15}$ | $A$ | -3.009*** | 0.128 | −23.481 |
| $\beta_{16}$ | $I$ | -15.932*** | 0.635 | −25.075 |
| $\beta_{17}$ | $W$ | -18.960*** | 0.892 | −21.250 |
| $\beta_{18}$ | $F$ | 0.157 | 0.216 | 0.728 |
| $\beta_{19}$ | $H$ | 86.502*** | 5.285 | 16.367 |
| $\beta_{20}$ | $D$ | 62.935*** | 1.934 | 32.540 |
| $\sigma^2$ | | 0.151*** | 0.031 | 4.920 |
| $\gamma$ | | 0.851*** | 0.065 | 13.126 |
| Log likelihood function value | | 16.228 | | |
| LR test | | 21.999 | | |

Note: *** indicate significance at the 1% levels, respectively.

and likelihood-ratio (LR) tests in the model estimation results are both significant at the 1% level of significance, indicating that it is feasible to choose the stochastic frontier analysis method for efficiency assessment. At the same time, the r value in the model is 0.851, indicating that the difference between the actual output and the ideal output of citrus fruit production is mainly caused by technical inefficiency, and the combination of these three indicators shows that the regression results are better and the model has applicability to the sample.

According to the estimation results of the stochastic frontier function, the coefficient of material and service input (lnK) is positive and passes the 1% significance test, indicating it is a core driver of citrus fruit output. An increase in these inputs significantly boosts production.The cross-term between time and direct material inputs (tlnK) is significantly positive, indicating that the yield-enhancing effects of direct inputs such as agricultural supplies and equipment continue to strengthen over time, with the benefits of production technology iteration gradually becoming apparent.The coefficient for labor input (lnL) is insignificant and negative. Combined with the significant negative result of the cross-term between time and labor (tlnL), this indicates inefficient redundancy in traditional labor within citrus planting. The inadequate skill alignment of this labor pool inhibits output growth.The cross-term between labor and indirect material inputs (lnLlnS) is significantly positive, indicating that indirect inputs such as technical services must synergize with labor operations to effectively

boost production.The cross-term between capital and indirect material inputs (lnKlnS) is significantly negative, indicating insufficient structural matching between direct and indirect material inputs. This may stem from incompatibility between capital inputs (e.g., equipment) and other inputs (e.g., raw materials), suggesting antagonistic effects that necessitate adjustments to the input structure.

The parameter estimates for factors affecting inefficiency reveal that the regional variable (A) is significantly negative at the 1% level. This indicates substantial geographical disparities in citrus cultivation technical efficiency, with eastern regions demonstrating markedly superior technical efficiency compared to central and western regions. The core reason lies in the gap between regional development foundations and technology dissemination levels. Per capita regional GDP (I) is significantly negative at the 1% level, not unrelated to technological efficiency. Rather, it stems from higher land and labor costs in economically developed regions, which diminish the comparative advantage of citrus planting. Resources consequently shift toward higher-yielding industries, indirectly leading to reduced production efficiency. The level of hired labor (W) is significantly negative at the 1% significance level, indicating that an increase in the labor hire rate reduces technical efficiency. This reflects inefficient redundancy in traditional labor practices within citrus planting, where overreliance on manual labor rather than mechanized operations diminishes technical efficiency. Furthermore, since the level of hired labor represents farm scale, this suggests that in citrus planting, excessive scale actually lowers technical efficiency. Financial support for agriculture (F) failed to pass the significance test, indicating that the current financial support for agriculture has not yet fully demonstrated its role in promoting the technical efficiency of citrus planting. The allocation or utilization efficiency of these funds requires optimization. The road network density (H) is significantly positive at the 1% level, indicating that increased road network density enhances regional accessibility, reduces transportation costs for agricultural inputs and product sales, and significantly boosts citrus planting efficiency. The incidence rate of meteorological disasters (D) passed the 1% significance test, indicating that meteorological disasters exert a certain impact on citrus planting. The higher the proportion of agricultural natural disasters, the lower the technical efficiency of citrus planting.

## Technical efficiency analysis of citrus planting

According to the calculation method of formula (3), the technical efficiency of citrus planting in the main production provinces from 2012 to 2024 was determined, as shown in Table 3. Overall, the technical efficiency of citrus fruit production exhibits a development trend of growth followed by a slow decline, ultimately stabilizing. Guangdong, Fujian, and Chongqing are the top three among the seven main producing provinces in terms of average technical efficiency of citrus planting, with averages of 0.938, 0.946, and 0.916, respectively. The average technical efficiency of citrus planting in Guangdong Province reached its highest value of 0.972 in 2024. In Fujian Province, the highest value was 0.964 in 2018, and in Chongqing Municipality, the highest value was 0.964 in 2017. Hubei, Hunan, and Jiangxi provinces are among the bottom three provinces in terms of average technical efficiency in citrus planting, and the technical efficiency of planting in these provinces still needs further improvement.

Table 3. Technical efficiency of citrus planting in 7 sample provinces and cities, 2012–2024.

| Provinces and municipalities | 2012 | 2013 | 2014 | 2015 | 2016 | 2017 | 2018 | 2019 | 2020 | 2021 | 2022 | 2023 | 2024 |
|---|---|---|---|---|---|---|---|---|---|---|---|---|---|
| Fujian | 0.93 | 0.882 | 0.948 | 0.952 | 0.948 | 0.946 | 0.964 | 0.936 | 0.94 | 0.96 | 0.937 | 0.893 | 0.956 |
| Guangdong | 0.933 | 0.928 | 0.938 | 0.87 | 0.959 | 0.937 | 0.965 | 0.962 | 0.959 | 0.971 | 0.958 | 0.943 | 0.972 |
| Hubei | 0.539 | 0.515 | 0.898 | 0.623 | 0.596 | 0.742 | 0.871 | 0.892 | 0.841 | 0.89 | 0.898 | 0.849 | 0.95 |
| Hunan | 0.402 | 0.367 | 0.712 | 0.893 | 0.824 | 0.697 | 0.767 | 0.899 | 0.811 | 0.92 | 0.628 | 0.934 | 0.923 |
| Jiangxi | 0.256 | 0.339 | 0.804 | 0.738 | 0.722 | 0.836 | 0.945 | 0.91 | 0.906 | 0.622 | 0.924 | 0.94 | 0.772 |
| Chongqing | 0.769 | 0.843 | 0.89 | 0.923 | 0.95 | 0.964 | 0.935 | 0.935 | 0.935 | 0.955 | 0.917 | 0.944 | 0.947 |
| Guangxi | 0.614 | 0.728 | 0.87 | 0.885 | 0.657 | 0.796 | 0.731 | 0.865 | 0.778 | 0.722 | 0.726 | 0.89 | 0.955 |

Fig 2 shows the value and trend of the average production technical efficiency of citrus from 2012 to 2024, from which the following conclusion can be drawn: in general, in the past 10 years, the average technical efficiency of citrus planting has followed a development pattern characterized by rapid growth followed by decline, then growth followed by decline, and finally stabilizing. From 2012 to 2014, the technical efficiency experienced the most significant fluctuations. Between 2014 and 2024, the technical efficiency fluctuated within a narrow range of 0.866 to 0.925, exhibiting a pattern of initial decline followed by growth, then another decline before rising again. It reached its peak in 2019, demonstrating a marked improvement in technical efficiency. Subsequently, it showed a downward trend before gradually stabilizing.

As can be seen in Fig 3, during the period 2012–2024, there are regional differences in citrus planting. The average technical efficiency of the eastern region is significantly higher than that of the central and western regions, and the average technical efficiency of the western region is higher than that of the central region.

The overall technical efficiency of citrus planting in the eastern region has shown a stable development trend. In the western region, citrus planting technical efficiency has exhibited fluctuating growth. The central region has experienced a pattern of significant growth followed by decline and subsequent recovery. From 2018 to 2024, the average technical efficiency in both the central and western regions demonstrated convergent trends, initially rising modestly before gradually declining and eventually stabilizing.

## Results

Based on panel data from seven major citrus planting regions spanning 2012–2024, this study adopts stochastic frontier production functions to analyze their technical efficiency. Results indicate that direct material inputs (e.g., fertilizers, agricultural machinery) are core drivers of citrus output, yet their marginal returns diminish with increasing input scale. Concurrently, traditional labor exhibits inefficiencies and redundancy, with skills failing to align with technological upgrades. Conversely, synergies between new agricultural inputs and time, alongside coordinated labor and technical services, significantly enhance output. At the level of technical efficiency, the average efficiency of citrus planting reached 0.837, with significant regional variations—eastern regions generally outperforming central and western areas. Enhancements in transportation infrastructure and logistics services can substantially boost efficiency, while excessive labor hiring and resource tilting toward economically developed regions can drag down efficiency. The impact of current financial support

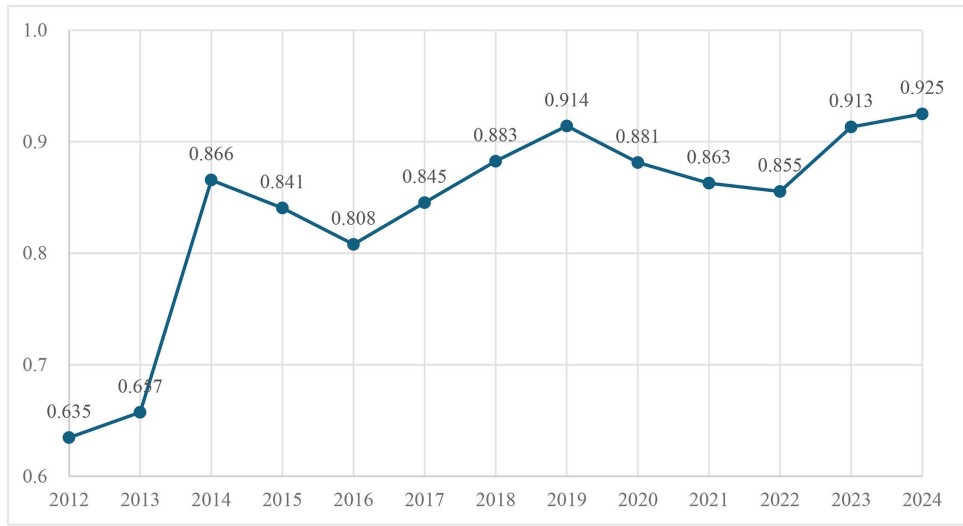

**Fig 2. Trends in average technical efficiency of citrus planting, 2012–2024.**

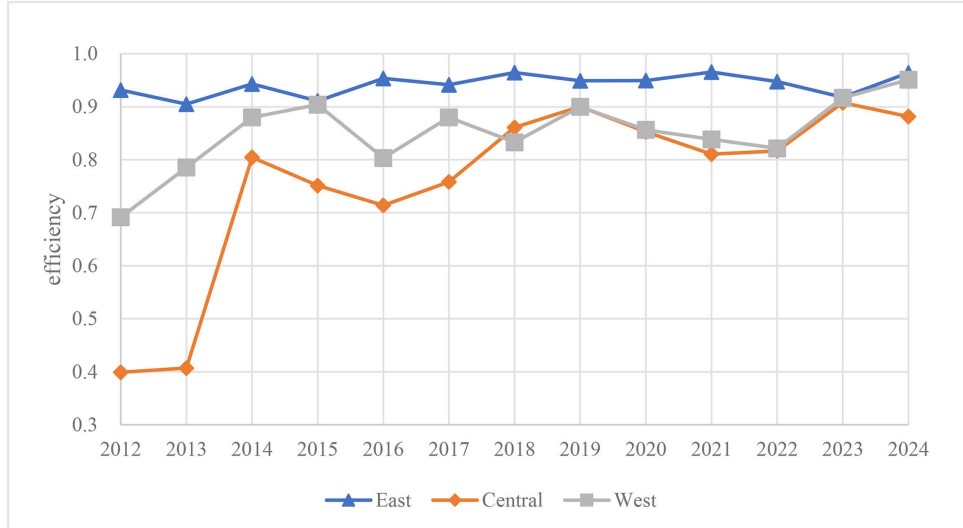

**Fig 3. Trends in average technical efficiency of citrus planting by region, 2012–2024.**

for agriculture has yet to materialize. Overall, while citrus planting efficiency remains relatively high, there is room for optimizing factor allocation. Further efficiency gains can be achieved through mechanization to replace redundant labor, region-specific support measures, and targeted fiscal investments.

## Discussion

Technological progress plays an important role in citrus planting, relying solely on the growth of inputs of fertilizers, pesticides, and labor does not bring about year-on-year growth in production, and even brings about a loss of efficiency from the point of view of efficiency, while the overuse of fertilizers and pesticides has a negative impact on the quality and safety of the citrus fruit produced. Therefore, in the process of citrus planting, it is necessary to harness the new growth momentum, the key to which is to increase investment in agricultural science and technology innovation. This will promote industrial innovation through scientific and technological advancements, ultimately enhancing technological efficiency. It is necessary to increase the normalization and standardization of citrus orchards and continually improve the level of citrus orchard planting and management, prioritize the research, development, and promotion of simplified, efficient, and practical cultivation techniques that are appropriate for specific locations and suitable for specific crop varieties. as well as advance the frontiers of production functions through technological progress. This is basically consistent with the findings of Fang Guozhu et al [3,14]. Therefore, the creation of standardized bases will be an important direction for future policy support.

The occurrence of natural disasters in agriculture has a significant negative impact on agricultural production, and affects the improvement of technical efficiency in citrus production. This is basically consistent with the research conclusions of Chen Xinjian et al [18]. In agricultural production, the monitoring for, and early warning of, natural disasters should be strengthened, especially by introducing modern information technology into agriculture, the establishment of the agro-meteorological disaster "sky-air-earth" integrated monitoring and early warning system, the research and development of agricultural natural disaster monitoring and early warning of new technologies, new methods, new products, and apply them to specific production practices, to enhance the ability to respond to disasters. Therefore, the use of digital technology empowerment to promote agricultural production efficiency will be an important direction of policy support.

The technical efficiency of citrus planting has regional differences, and in recent years, the technical efficiency of citrus panting in the western region has increased steadily, and the level of competitiveness of citrus planting in Chongqing, Guangxi, and other regions has been significantly improved, which indicates that the comparative advantages of the southwestern region in citrus production have been brought into play. However, citrus planting areas are mainly concentrated in mountainous areas, and citrus planting and management are mainly based on individual family units, consequently, the production scale is relatively small and the level of organization is low. The empirical research in this paper shows that the improvement of production scale can significantly improve the technical efficiency of production, so citrus planting in Southwest China needs to explore how to guide the effective transfer of citrus orchards, improve the scale of production and operation, achieve moderate scale operation, promote the improvement of technical efficiency, and then further improve the income level of citrus growers. This is basically consistent with the research conclusions of Zeng Linguo et al [17]. Therefore, guiding growers to moderate-scale cultivation is an important direction of existing policy support.

Due to the limitation of obtaining data, this study only considered the influence factors such as regional differences in citrus planting, level of hired labor, financial support to agriculture, density of highway network, and natural disasters in agriculture, and did not take into account factors such as temperature, soil, level of orchard mechanization, scientific and technological inputs, and education level of personnel engaged in citrus cultivation, etc., which affect citrus planting. Therefore, there are some limitations in measuring and analyzing the technical efficiency of citrus planting in the seven main citrus producing areas in China. Meanwhile, the stochastic frontier analysis model used in this paper can only measure technical efficiency values and differences in the main planting areas using the data. To provide more realistic countermeasure suggestions for formulating specific improvement strategies, it is necessary to conduct practical research in each region. In the future, environmental factors, mechanization levels, and other variables will be included in the model to better analyze the influencing factors affecting the technical efficiency of citrus planting, thereby continuously improving the technical efficiency of citrus planting and enhancing industrial competitiveness.

## Conclusion

This study uses planting and operation statistics from 2012 to 2024 for the seven main citrus-producing areas: Fujian, Guangdong, Hubei, Jiangxi, Hunan, Chongqing, and Guangxi. It adopts the beyond the logarithmic stochastic frontier analysis method to study the cultivation technical efficiency of citrus fruit. The factors affecting production technical efficiency are analyzed, and four main conclusions are drawn from the empirical results.

(1) During the period studied, the overall technical efficiency of citrus cultivation in China's major citrus-producing provinces exhibited a developmental trajectory characterized by rapid growth followed by a gradual decline, then slow growth before another decline, and finally stabilizing.In terms of horizontal comparison among the sample provinces, citrus planting in Guangdong Province had the highest technical efficiency, which tended to stabilize, while citrus planting in Jiangxi Province had the lowest average technical efficiency, Hunan, Hubei, and Guangxi have comparable levels of technical efficiency in citrus planting.

(2) In citrus planting, the effects of increasing labor inputs, direct material and service inputs (mainly in fertilizers and pesticides), and indirect material and service inputs (mainly in depreciation of fixed assets and sales fees on output) are positive, i.e., increases in the first three inputs positively contribute to an increase in citrus output, but citrus output declines over time provided that labor and capital inputs remain constant.

(3) There are regional differences in the technical efficiency of citrus planting. Eastern regions demonstrate significantly higher technical efficiency in citrus planting compared to central and western regions, yet the growth rate of technical efficiency in central and western regions markedly exceeds that of the eastern region. Since 2018, the overall changes in average technical efficiency across central and western regions have shown a tendency toward convergence.

(4) In the factors affecting the technical efficiency of citrus planting, the increase in the level of hiring laborers can promote the technical efficiency of citrus planting, i.e., by expanding the planting. However, if the scale is too large, the technical efficiency of planting will be reduced. The increase of road network density has a positive effect on the promotion of technical efficiency of citrus planting, i.e., the higher the density of the road network, the greater the technical efficiency of citrus planting. The occurrence of meteorological disasters will have a negative impact on citrus production, and the higher the ratio of natural disasters in agriculture, the lower the technical efficiency of citrus planting.

## Supporting information

**S1 Dataset. Original data used in the paper (Output value per mu, Labor input per mu, Direct material and service input per mu, Indirect material and service input per mu, Region, Employment rate, Gross regional product per capita, Financial support for agriculture, Road network density, Disaster rate).**
(XLSX)

**S1 Table. Frontier 4.1 software run results(Stochastic frontier function estimation results).**
(DOCX)

**S1 Fig. Trends in average technical efficiency of citrus planting, 2012–2024.** Trends in average technical efficiency of citrus planting by region, 2012–2024.
(XLSX)

## Author contributions

**Data curation:** Xiaojun Pu.

**Methodology:** yuan wu.

**Writing – original draft:** yuan wu.

**Writing – review & editing:** yuan wu.

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
