## [Decision Letter · Decision Letter 0]

4 Apr 2025

PONE-D-25-04501Evaluating the Technical Efficiency and Influencing Factors of Citrus Fruits Planting in ChinaPLOS ONE

Dear Dr. wu,

Thank you for submitting your manuscript to PLOS ONE. After careful consideration, we feel that it has merit but does not fully meet PLOS ONE’s publication criteria as it currently stands. Therefore, we invite you to submit a revised version of the manuscript that addresses the points raised during the review process.

We look forward to receiving your revised manuscript.

Kind regards,

Noé Aguilar-Rivera

Academic Editor

PLOS ONE

Journal Requirements:

2. PLOS ONE publication criteria requires that research must be described in enough detail to allow readers to fully replicate the study (https://journals.plos.org/plosone/s/criteria-for-publication#loc-3), and that all data underlying the findings described in their manuscript must be fully available (https://journals.plos.org/plosone/s/criteria-for-publication#loc-331). Please amend your Methods section to provide the URL for the specific data sets used and please provide the datasets as a supplemental file. In addition please clarify if your dataset age is 2012–2022, as mentioned ion your abstract, or 2013–2023, as stated in your methods section.

Reviewers' comments:

Reviewer's Responses to Questions

**Comments to the Author**

1. Is the manuscript technically sound, and do the data support the conclusions?

Reviewer #1: Yes

Reviewer #2: Partly

2. Has the statistical analysis been performed appropriately and rigorously? 

Reviewer #1: Yes

Reviewer #2: N/A

3. Have the authors made all data underlying the findings in their manuscript fully available?

The PLOS Data policy requires authors to make all data underlying the findings described in their manuscript fully available without restriction, with rare exception (please refer to the Data Availability Statement in the manuscript PDF file). The data should be provided as part of the manuscript or its supporting information, or deposited to a public repository. For example, in addition to summary statistics, the data points behind means, medians and variance measures should be available. If there are restrictions on publicly sharing data—e.g. participant privacy or use of data from a third party—those must be specified.requires authors to make all data underlying the findings described in their manuscript fully available without restriction, with rare exception (please refer to the Data Availability Statement in the manuscript PDF file). The data should be provided as part of the manuscript or its supporting information, or deposited to a public repository. For example, in addition to summary statistics, the data points behind means, medians and variance measures should be available. If there are restrictions on publicly sharing data—e.g. participant privacy or use of data from a third party—those must be specified.requires authors to make all data underlying the findings described in their manuscript fully available without restriction, with rare exception (please refer to the Data Availability Statement in the manuscript PDF file). The data should be provided as part of the manuscript or its supporting information, or deposited to a public repository. For example, in addition to summary statistics, the data points behind means, medians and variance measures should be available. If there are restrictions on publicly sharing data—e.g. participant privacy or use of data from a third party—those must be specified.requires authors to make all data underlying the findings described in their manuscript fully available without restriction, with rare exception (please refer to the Data Availability Statement in the manuscript PDF file). The data should be provided as part of the manuscript or its supporting information, or deposited to a public repository. For example, in addition to summary statistics, the data points behind means, medians and variance measures should be available. If there are restrictions on publicly sharing data—e.g. participant privacy or use of data from a third party—those must be specified.

Reviewer #1: No

Reviewer #2: Yes

4. Is the manuscript presented in an intelligible fashion and written in standard English?

Reviewer #1: Yes

Reviewer #2: No

5. Review Comments to the Author

Reviewer #1: Review of "Evaluating the Technical Efficiency and Influencing Factors of Citrus Fruits Planting in China"

This paper presents an interesting study on the technical efficiency of citrus fruit planting in China. The authors employ a stochastic frontier analysis (SFA) model with a beyond logarithmic production function to analyze panel data from seven major citrus-producing provinces and cities in China from 2012 to 2022. The topic is relevant, particularly given the importance of the citrus industry in China and the need to improve its production efficiency. The methodology is generally sound, and the results provide some useful insights. However, there are some weaknesses that need to be addressed before the manuscript can be considered for publication.

Weaknesses:

Justification of Variables: While the authors explain the variables used in the model, a more in-depth justification for their inclusion and potential impact would be beneficial. For example, the choice of "indirect material and service input" could be further elaborated.

Discussion of Limitations: The authors should include a more thorough discussion of the limitations of their study. This could include potential biases in the data, limitations of the model, and the generalizability of the results.

Specific Comments and Suggestions for Revision:

1. Introduction:

The introduction provides a good overview of the citrus industry in China. However, it could benefit from a clearer statement of the research gap and the specific contributions of this study.

2. Literature Review:

The literature review provides a good overview of previous studies on efficiency analysis in agriculture, particularly in the citrus sector. However, it could be improved by:

Explicitly stating how this study builds upon or differs from previous research.

3. Model Construction:

The model construction section is generally well-explained. However:

The justification for including specific variables in the inefficiency function could be strengthened.

4. Results and Discussion:

The results are presented clearly, and the discussion is generally good. However:

The discussion could be more in-depth, providing more context for the findings and comparing them with previous research.

The authors should discuss the policy implications of their findings in more detail.

5. Conclusions:

The conclusions summarize the main findings of the study. However:

The authors should reiterate the limitations of the study and suggest directions for future research.

Reviewer #2: -The manuscript should be structured to include the following sections: an Introduction, which provides background information and outlines the study's objectives; Materials and Methods, detailing the experimental design, data collection, and analytical techniques; Results, presenting the key findings objectively; and Discussion, interpreting the results in the context of existing literature and highlighting their implications.

-The manuscript requires revision to improve the quality of the English language."

-Key words: Aren’t keywords usually supposed to be different from words already mentioned in title and abstract?

-The introduction lacks appropriate references to support key statements and provide context for the study.

-Why has the author separated the Introduction and Literature Review into distinct sections?

-The line spacing format is inconsistent.

-The citation format in the text is inconsistent, as the author alternates between using numbered references and author names for citations.

-Additionally, the citation style should include only the first author’s name, followed by 'et al.' when citing multiple authors.

-What does the author mean by the term “natural disasters in agriculture”?

-Huanglongbing (HLB) is an important factor affecting citrus production efficiency in China, and its inclusion in the study will significantly enhance this finding. It is highly recommended to include this factor in the study. Also, the study does not sufficiently explore sustainable alternatives such as organic farming, integrated pest management (IPM), or biofertilizers on citrus production in China.

-The author did not provide sufficient justification or supporting evidence for conducting this research.

-The conclusion is overly lengthy and could be more concise.

6. PLOS authors have the option to publish the peer review history of their article (what does this mean?). If published, this will include your full peer review and any attached files.). If published, this will include your full peer review and any attached files.). If published, this will include your full peer review and any attached files.). If published, this will include your full peer review and any attached files.

...

Reviewer #1: No

Reviewer #2: **Yes:** Saoussen Ben AbdallahSaoussen Ben AbdallahSaoussen Ben AbdallahSaoussen Ben Abdallah

---

## [Author Response · Author response to Decision Letter 1]

24 May 2025

Review of "Evaluating the Technical Efficiency and Influencing Factors of Citrus Fruits Planting in China"

This paper presents an interesting study on the technical efficiency of citrus fruit planting in China. The authors employ a stochastic frontier analysis (SFA) model with a beyond logarithmic production function to analyze panel data from seven major citrus-producing provinces and cities in China from 2012 to 2022. The topic is relevant, particularly given the importance of the citrus industry in China and the need to improve its production efficiency. The methodology is generally sound, and the results provide some useful insights. However, there are some weaknesses that need to be addressed before the manuscript can be considered for publication.

Weaknesses:

• Justification of Variables: While the authors explain the variables used in the model, a more in-depth justification for their inclusion and potential impact would be beneficial. For example, the choice of "indirect material and service input" could be further elaborated.

• The rationale for the selection of these variables was mainly based on the actual situation of citrus planting, the combination of previous relevant studies, and the availability of data, which has been described in the text.

• Discussion of Limitations: The authors should include a more thorough discussion of the limitations of their study. This could include potential biases in the data, limitations of the model, and the generalizability of the results.

• The limitations of the study are explained. See the “Discussion” section for details.

Specific Comments and Suggestions for Revision:

1. Introduction:

o The introduction provides a good overview of the citrus industry in China. However, it could benefit from a clearer statement of the research gap and the specific contributions of this study.

o Changes have been made, as requested, to add the specific contributions of the study, as detailed at the end of the first paragraph of the Introduction.

2. Literature Review:

o The literature review provides a good overview of previous studies on efficiency analysis in agriculture, particularly in the citrus sector. However, it could be improved by:

Explicitly stating how this study builds upon or differs from previous research.

Changes have been made, as requested, to add the specific contributions of the study, as detailed at the end of the first paragraph of the Introduction.

3. Model Construction:

o The model construction section is generally well-explained. However:

The justification for including specific variables in the inefficiency function could be strengthened.

The specific variables included in the inefficiency function are based on the actual production situation, previous studies, and data availability, as described in the “Model Construction” section.

4. Results and Discussion:

o The results are presented clearly, and the discussion is generally good. However:

The discussion could be more in-depth, providing more context for the findings and comparing them with previous research.

The authors should discuss the policy implications of their findings in more detail.

Comparisons with previous studies are made and possible policy implications of the findings are described in the Discussion section.

5. Conclusions:

o The conclusions summarize the main findings of the study. However:

The authors should reiterate the limitations of the study and suggest directions for future research.

The limitations of the study are explained and future research directions are proposed, as detailed in the Discussion section.

Reviewer #2: -The manuscript should be structured to include the following sections: an Introduction, which provides background information and outlines the study's objectives; Materials and Methods, detailing the experimental design, data collection, and analytical techniques; Results, presenting the key findings objectively; and Discussion, interpreting the results in the context of existing literature and highlighting their implications.

The manuscript has been revised according to the structure of “Introduction, Materials and Methods, Results, Discussion”.

-The manuscript requires revision to improve the quality of the English language."

-Key words: Aren’t keywords usually supposed to be different from words already mentioned in title and abstract?

Keywords have been deleted.

-The introduction lacks appropriate references to support key statements and provide context for the study.

The structure of the article has been restructured, and there is a review and introduction of relevant research in the context of the study.

-Why has the author separated the Introduction and Literature Review into distinct sections?

The article structure was restructured, and the introduction and article review were merged.

-The line spacing format is inconsistent.

Adjusted to harmonized spacing.

-The citation format in the text is inconsistent, as the author alternates between using numbered references and author names for citations.

-Additionally, the citation style should include only the first author’s name, followed by 'et al.' when citing multiple authors.

The citation format of the references has been revised as required.

-What does the author mean by the term “natural disasters in agriculture”?

Natural disasters in agriculture mainly refer to floods, droughts, etc.

-Huanglongbing (HLB) is an important factor affecting citrus production efficiency in China, and its inclusion in the study will significantly enhance this finding. It is highly recommended to include this factor in the study. Also, the study does not sufficiently explore sustainable alternatives such as organic farming, integrated pest management (IPM), or biofertilizers on citrus production in China.

-The author did not provide sufficient justification or supporting evidence for conducting this research.

-The conclusion is overly lengthy and could be more concise.

Huanglongbing (HLB) is an important factor affecting citrus production efficiency in China。However, it was not included in the analysis of influencing factors in this paper due to some difficulties in data collection and acquisition, which could be included in subsequent studies.

In addition, the structure and content of the article has been reorganized, splitting the original conclusion into two parts, conclusion and discussion, which makes a bit more sense.

---

## [Decision Letter · Decision Letter 1]

22 Jul 2025

PONE-D-25-04501R1Evaluating the Technical Efficiency and Influencing Factors of Citrus Fruits Planting in ChinaPLOS ONE

Dear Dr. wu,

Thank you for submitting your manuscript to PLOS ONE. After careful consideration, we feel that it has merit but does not fully meet PLOS ONE’s publication criteria as it currently stands. Therefore, we invite you to submit a revised version of the manuscript that addresses the points raised during the review process.

We look forward to receiving your revised manuscript.

Kind regards,

Noé Aguilar-Rivera

Academic Editor

PLOS ONE

Journal Requirements:

Reviewers' comments:

Reviewer's Responses to Questions

**Comments to the Author**

1. If the authors have adequately addressed your comments raised in a previous round of review and you feel that this manuscript is now acceptable for publication, you may indicate that here to bypass the “Comments to the Author” section, enter your conflict of interest statement in the “Confidential to Editor” section, and submit your "Accept" recommendation.

Reviewer #2: (No Response)

Reviewer #3: All comments have been addressed

2. Is the manuscript technically sound, and do the data support the conclusions?

Reviewer #2: Partly

Reviewer #3: Partly

3. Has the statistical analysis been performed appropriately and rigorously? 

Reviewer #2: I Don't Know

Reviewer #3: Yes

4. Have the authors made all data underlying the findings in their manuscript fully available?

The PLOS Data policy requires authors to make all data underlying the findings described in their manuscript fully available without restriction, with rare exception (please refer to the Data Availability Statement in the manuscript PDF file). The data should be provided as part of the manuscript or its supporting information, or deposited to a public repository. For example, in addition to summary statistics, the data points behind means, medians and variance measures should be available. If there are restrictions on publicly sharing data—e.g. participant privacy or use of data from a third party—those must be specified.requires authors to make all data underlying the findings described in their manuscript fully available without restriction, with rare exception (please refer to the Data Availability Statement in the manuscript PDF file). The data should be provided as part of the manuscript or its supporting information, or deposited to a public repository. For example, in addition to summary statistics, the data points behind means, medians and variance measures should be available. If there are restrictions on publicly sharing data—e.g. participant privacy or use of data from a third party—those must be specified.requires authors to make all data underlying the findings described in their manuscript fully available without restriction, with rare exception (please refer to the Data Availability Statement in the manuscript PDF file). The data should be provided as part of the manuscript or its supporting information, or deposited to a public repository. For example, in addition to summary statistics, the data points behind means, medians and variance measures should be available. If there are restrictions on publicly sharing data—e.g. participant privacy or use of data from a third party—those must be specified.requires authors to make all data underlying the findings described in their manuscript fully available without restriction, with rare exception (please refer to the Data Availability Statement in the manuscript PDF file). The data should be provided as part of the manuscript or its supporting information, or deposited to a public repository. For example, in addition to summary statistics, the data points behind means, medians and variance measures should be available. If there are restrictions on publicly sharing data—e.g. participant privacy or use of data from a third party—those must be specified.

Reviewer #2: Yes

Reviewer #3: Yes

5. Is the manuscript presented in an intelligible fashion and written in standard English?

Reviewer #2: No

Reviewer #3: Yes

6. Review Comments to the Author

Reviewer #2: I have a few suggestions and comments to improve the clarity, accuracy, and structure of the manuscript: Introduction:

-The study is being submitted in 2025, yet the Introduction heavily relies on data from 2022. Please consider updating your citrus production statistics (both globally and for China) to reflect the most recent data available.

-The Introduction would benefit from additional, more recent references, particularly when discussing global and national citrus production trends and the importance of citrus in China's agricultural economy.

-The current Introduction dives directly into citrus importance and efficiency measurements without first framing the broader challenges facing citrus production (e.g., climate change, pest pressure, resource inefficiency, market access). I suggest starting with a short paragraph that clearly outlines the general problems in citrus production, and then narrowing the focus to technical efficiency, highlighting previous studies and where the research gap lies.

-Please ensure that the in-text citation format is consistent throughout the manuscript. Currently, the text switches between numbered references and author-date narrative formats (e.g., "Xiong Wei et al." vs. [8]). Align with the journal’s required citation style.

-In the section where the objectives of the study are stated, please clearly mention the seven provinces/cities included in your panel data analysis. This will help readers immediately understand the geographical scope and relevance of the study.

Material and methods

Many parts of the material and methods need to be moved to the introduction section.

For example:

“Citrus planting is a typical labor-intensive task [22] , and from a practical point of view, citrus harvests are affected by labor intensity, fertilizer and pesticide inputs, etc., based on the current situation and the caliber of agricultural statistics and data availability…”.

“The production activities of citrus planting are profoundly affected by the impacts of the natural and social environments. For example, climate has an important impact on citrus planting[24]. Citrus frost damage not only affects plant production but also has a sustained impact on production over a number of years, and the problem of transport accessibility in citrus-producing areas has a direct impact on whether citrus crops can be marketed 6 successfully[25].”

The data source

The first paragraph of data sources should be moved to the introduction for objective goals.

The model estimation and analysis

This section presents the core findings of your model estimation and variable significance testing. I strongly recommend incorporating this section into the Results section. This would improve the organization of your manuscript and help the reader follow your analytical progression.

Discussion

The Discussion section requires stronger connections with your model results, more nuanced interpretation, and clearer implications for practice and policy. I encourage you to deepen the analysis and focus on evidence-based recommendations. These improvements will significantly enhance the scientific contribution of your work.

-The English language and writing quality need improvement to enhance clarity and academic readability.

Reviewer #3: Abstract: Ok. The last three paragraphs of the summary are not significant to the text.

Introduction: Use more up-to-date statistics. In 2023, global citrus production was 217 million tons. Data from FAOSTAT https://www.fao.org/faostat/es/#data/QCL Also, data from China, reporting 95 million tons in 2023. Update performance data, in 2023 it was 14,800 kg/hm2, compared to Paraguay, Indonesia or South Africa which is three times higher. There are no DEA studies that link production with environmental or soil aspects? Are there any studies on the productivity of countries with high citrus yields?

Materials and methods: Social data can be very subjective, so how do you handle qualitative data? Also, when using previous studies, they should be from locations like the one in this study. It is suggested that a location map of the study area be added. When using scales in plot size, indicate that it represents large or small scales in a unit of measurement. During the analysis period, associate data with regional climate phenomena.

Results: There is information found in materials and methods that could be reported in results. Adding a graph or map by region or zone would be helpful for comparing information.

Discussion: The discussion could be supported by a graph to reinforce what is stated in the text. A basic analysis of the information is conducted; the data used in the models could help foster further discussion, given the time and space.

No conclusions to the work?

7. PLOS authors have the option to publish the peer review history of their article (what does this mean?). If published, this will include your full peer review and any attached files.). If published, this will include your full peer review and any attached files.). If published, this will include your full peer review and any attached files.). If published, this will include your full peer review and any attached files.

...

Reviewer #2: **Yes:** Saoussen Ben AbdallahSaoussen Ben AbdallahSaoussen Ben AbdallahSaoussen Ben Abdallah

Reviewer #3: **Yes:** LUIS ALBERTO OLVERA-VARGASLUIS ALBERTO OLVERA-VARGASLUIS ALBERTO OLVERA-VARGASLUIS ALBERTO OLVERA-VARGAS

---

## [Author Response · Author response to Decision Letter 2]

1 Sep 2025

Reviewer #2: I have a few suggestions and comments to improve the clarity, accuracy, and structure of the manuscript:

Introduction:

-The study is being submitted in 2025, yet the Introduction heavily relies on data from 2022. Please consider updating your citrus production statistics (both globally and for China) to reflect the most recent data available.

The data has been updated to the latest version as requested. Please refer to the introduction section for details.

-The Introduction would benefit from additional, more recent references, particularly when discussing global and national citrus production trends and the importance of citrus in China's agricultural economy.

-The current Introduction dives directly into citrus importance and efficiency measurements without first framing the broader challenges facing citrus production (e.g., climate change, pest pressure, resource inefficiency, market access). I suggest starting with a short paragraph that clearly outlines the general problems in citrus production, and then narrowing the focus to technical efficiency, highlighting previous studies and where the research gap lies.

The introduction section has been appropriately edited to include common issues faced in citrus planting.

-Please ensure that the in-text citation format is consistent throughout the manuscript. Currently, the text switches between numbered references and author-date narrative formats (e.g., "Xiong Wei et al." vs. [8]). Align with the journal’s required citation style.

The revisions have been made as requested. The citation format is consistent with that of the journal's reference list.

-In the section where the objectives of the study are stated, please clearly mention the seven provinces/cities included in your panel data analysis. This will help readers immediately understand the geographical scope and relevance of the study.

Seven major citrus growing regions were added at the end of the introduction.

Material and methods

Many parts of the material and methods need to be moved to the introduction section.

For example:

“Citrus planting is a typical labor-intensive task [22] , and from a practical point of view, citrus harvests are affected by labor intensity, fertilizer and pesticide inputs, etc., based on the current situation and the caliber of agricultural statistics and data availability…”.

“The production activities of citrus planting are profoundly affected by the impacts of the natural and social environments. For example, climate has an important impact on citrus planting[24]. Citrus frost damage not only affects plant production but also has a sustained impact on production over a number of years, and the problem of transport accessibility in citrus-producing areas has a direct impact on whether citrus crops can be marketed 6 successfully[25].”

This section mainly aims to objectively describe factors that may affect the inefficiency of citrus production, thereby incorporating regional economic differences, climatic differences, transportation infrastructure conditions, etc. into the analytical framework to construct an inefficiency function.

The data source

The first paragraph of data sources should be moved to the introduction for objective goals.

This section provides a brief overview of why these seven regions were selected as samples for studying citrus efficiency issues.

The model estimation and analysis

This section presents the core findings of your model estimation and variable significance testing. I strongly recommend incorporating this section into the Results section. This would improve the organization of your manuscript and help the reader follow your analytical progression.

Discussion

The Discussion section requires stronger connections with your model results, more nuanced interpretation, and clearer implications for practice and policy. I encourage you to deepen the analysis and focus on evidence-based recommendations. These improvements will significantly enhance the scientific contribution of your work.

Modifications have been made as requested.

-The English language and writing quality need improvement to enhance clarity and academic readability.

Reviewer #3: Abstract: Ok. The last three paragraphs of the summary are not significant to the text.

Introduction: Use more up-to-date statistics. In 2023, global citrus production was 217 million tons. Data from FAOSTAT https://www.fao.org/faostat/es/#data/QCL Also, data from China, reporting 95 million tons in 2023. Update performance data, in 2023 it was 14,800 kg/hm2, compared to Paraguay, Indonesia or South Africa which is three times higher. There are no DEA studies that link production with environmental or soil aspects? Are there any studies on the productivity of countries with high citrus yields?

The relevant data has been updated in accordance with the opinion.

Materials and methods: Social data can be very subjective, so how do you handle qualitative data? Also, when using previous studies, they should be from locations like the one in this study. It is suggested that a location map of the study area be added. When using scales in plot size, indicate that it represents large or small scales in a unit of measurement. During the analysis period, associate data with regional climate phenomena.

Based on actual production conditions, combined with previous relevant research and data availability, regional economic differences, climate differences, transportation infrastructure conditions, and other factors are incorporated into the analytical framework to objectively describe factors that may affect the inefficiency of citrus production and construct an inefficiency function.

Results: There is information found in materials and methods that could be reported in results. Adding a graph or map by region or zone would be helpful for comparing information.

The relevant results obtained in the article have been summarized in the conclusion section.

Discussion: The discussion could be supported by a graph to reinforce what is stated in the text. A basic analysis of the information is conducted; the data used in the models could help foster further discussion, given the time and space.

No conclusions to the work?

Figures have been added to the text to clearly illustrate the basic situation of citrus production value in the main citrus-producing regions. The conclusion section has been expanded.

---

## [Decision Letter · Decision Letter 2]

29 Oct 2025

PONE-D-25-04501R2

Evaluating the Technical Efficiency and Influencing Factors of Citrus Fruits Planting in China

PLOS ONE

Dear Dr. wu,

Thank you for submitting your manuscript to PLOS ONE. After careful consideration, we feel that it has merit but does not fully meet PLOS ONE’s publication criteria as it currently stands. Therefore, we invite you to submit a revised version of the manuscript that addresses the points raised during the review process.

We look forward to receiving your revised manuscript.

Kind regards,

Noé Aguilar-Rivera

Academic Editor

PLOS ONE

Journal Requirements:

Reviewers' comments:

Reviewer's Responses to Questions

**Comments to the Author**

1. If the authors have adequately addressed your comments raised in a previous round of review and you feel that this manuscript is now acceptable for publication, you may indicate that here to bypass the “Comments to the Author” section, enter your conflict of interest statement in the “Confidential to Editor” section, and submit your "Accept" recommendation.

Reviewer #2: (No Response)

Reviewer #3: All comments have been addressed

2. Is the manuscript technically sound, and do the data support the conclusions?

Reviewer #2: Yes

Reviewer #3: Yes

3. Has the statistical analysis been performed appropriately and rigorously?

Reviewer #2: I Don't Know

Reviewer #3: Yes

4. Have the authors made all data underlying the findings in their manuscript fully available?

The PLOS Data policy requires authors to make all data underlying the findings described in their manuscript fully available without restriction, with rare exception (please refer to the Data Availability Statement in the manuscript PDF file). The data should be provided as part of the manuscript or its supporting information, or deposited to a public repository. For example, in addition to summary statistics, the data points behind means, medians and variance measures should be available. If there are restrictions on publicly sharing data—e.g. participant privacy or use of data from a third party—those must be specified. requires authors to make all data underlying the findings described in their manuscript fully available without restriction, with rare exception (please refer to the Data Availability Statement in the manuscript PDF file). The data should be provided as part of the manuscript or its supporting information, or deposited to a public repository. For example, in addition to summary statistics, the data points behind means, medians and variance measures should be available. If there are restrictions on publicly sharing data—e.g. participant privacy or use of data from a third party—those must be specified. requires authors to make all data underlying the findings described in their manuscript fully available without restriction, with rare exception (please refer to the Data Availability Statement in the manuscript PDF file). The data should be provided as part of the manuscript or its supporting information, or deposited to a public repository. For example, in addition to summary statistics, the data points behind means, medians and variance measures should be available. If there are restrictions on publicly sharing data—e.g. participant privacy or use of data from a third party—those must be specified. requires authors to make all data underlying the findings described in their manuscript fully available without restriction, with rare exception (please refer to the Data Availability Statement in the manuscript PDF file). The data should be provided as part of the manuscript or its supporting information, or deposited to a public repository. For example, in addition to summary statistics, the data points behind means, medians and variance measures should be available. If there are restrictions on publicly sharing data—e.g. participant privacy or use of data from a third party—those must be specified.

Reviewer #2: Yes

Reviewer #3: Yes

5. Is the manuscript presented in an intelligible fashion and written in standard English?

Reviewer #2: No

Reviewer #3: Yes

6. Review Comments to the Author

Reviewer #2: Reviewer Comment:

The manuscript has been notably improved; however, the English language still requires careful revision to enhance readability and clarity. The conclusion should be moved to the end of the manuscript to follow standard structure. Additionally, the quality and presentation of the tables should be improved.

Reviewer #3: A note: the conclusion is the final point of the manuscript. In the text, the conclusion was placed first and the discussion second. Change the order for better understanding.

7. PLOS authors have the option to publish the peer review history of their article (what does this mean?). If published, this will include your full peer review and any attached files.). If published, this will include your full peer review and any attached files.). If published, this will include your full peer review and any attached files.). If published, this will include your full peer review and any attached files.

...

Reviewer #2: **Yes:** Saoussen Ben AbdallahSaoussen Ben AbdallahSaoussen Ben AbdallahSaoussen Ben Abdallah

Reviewer #3: **Yes:** LUIS ALBERTO OLVERA VARGASLUIS ALBERTO OLVERA VARGASLUIS ALBERTO OLVERA VARGASLUIS ALBERTO OLVERA VARGAS

---

## [Author Response · Author response to Decision Letter 3]

23 Nov 2025

Reviewers' comments:

Reviewer's Responses to Questions

Comments to the Author

1. If the authors have adequately addressed your comments raised in a previous round of review and you feel that this manuscript is now acceptable for publication, you may indicate that here to bypass the “Comments to the Author” section, enter your conflict of interest statement in the “Confidential to Editor” section, and submit your "Accept" recommendation.

Reviewer #2: (No Response)

Reviewer #3: All comments have been addressed

2. Is the manuscript technically sound, and do the data support the conclusions?

Reviewer #2: Yes

Reviewer #3: Yes

3. Has the statistical analysis been performed appropriately and rigorously?

Reviewer #2: I Don't Know

Reviewer #3: Yes

Statistical analysis has been rigorously implemented.

4. Have the authors made all data underlying the findings in their manuscript fully available?

Reviewer #2: Yes

Reviewer #3: Yes

5. Is the manuscript presented in an intelligible fashion and written in standard English?

Reviewer #2: No

Reviewer #3: Yes

The language throughout the document has been refined. For details, please refer to the revised version.

6. Review Comments to the Author

Reviewer #2: Reviewer Comment:

The manuscript has been notably improved; however, the English language still requires careful revision to enhance readability and clarity. The conclusion should be moved to the end of the manuscript to follow standard structure. Additionally, the quality and presentation of the tables should be improved.

Reviewer #3: A note: the conclusion is the final point of the manuscript. In the text, the conclusion was placed first and the discussion second. Change the order for better understanding.

Thank you for the reviewers' comments. The language throughout the manuscript has been further revised. The conclusion section has been moved to the end of the paper. The formatting of the tables in the manuscript has been adjusted，please refer to the revised version for details.

7. PLOS authors have the option to publish the peer review history of their article (what does this mean?). If published, this will include your full peer review and any attached files.

Do you want your identity to be public for this peer review? For information about this choice, including consent withdrawal, please see our Privacy Policy.

Reviewer #2: Yes: Saoussen Ben Abdallah

Reviewer #3: Yes: LUIS ALBERTO OLVERA VARGAS

---

## [Decision Letter · Decision Letter 3]

30 Dec 2025

PONE-D-25-04501R3Evaluating the Technical Efficiency and Influencing Factors of Citrus Fruits Planting in ChinaPLOS One

Dear Dr. wu,

Thank you for submitting your manuscript to PLOS ONE. After careful consideration, we feel that it has merit but does not fully meet PLOS ONE’s publication criteria as it currently stands. Therefore, we invite you to submit a revised version of the manuscript that addresses the points raised during the review process.

**ACADEMIC EDITOR:**

6. Review Comments to the Author

Reviewer #2: Introduction

In the section beginning with “Citrus, indigenous to China … until the competitiveness of the industry”, no reference is provided to support the statements made. Appropriate citations should be added to substantiate this information.

Additionally, the authors report data up to 2023; however, given that we are now in 2025, it would strengthen the manuscript to include the most recent available data, along with corresponding references, to provide an updated perspective.

Regarding the citation of Chen Xinjian et al., the reference number for this study should be clearly indicated in the text to ensure consistency with the reference list.

It is recommended that the authors avoid using the phrase “this paper” and instead use “this study” or “this research”.

Material and methods

It would improve the clarity and logical flow of this section if the authors first describe the data sources, followed by the model construction used in the study.

Results

After the Materials and Methods section, the subheading Results is missing and should be added.

For Table 1, How the standard deviation of Indirect material and service input per mu (yuan) (279.24) exceeds the corresponding mean value (267.02)? The authors should explain this variability or verify the accuracy of the reported data.

In addition, no statistical analyses are reported for Table 1, Table 3, Figure 2, and Figure 3. Interpretation or discussion of differences and significance should only be presented if appropriate statistical tests were conducted. The authors are encouraged to include the relevant statistical analyses.

Discussion

The Discussion section needs to be improved. The authors should provide a deeper interpretation of their results, discussing it with previous studies. In particular, the findings could be better contextualized by comparing them with similar studies conducted in other countries, such as Brazil, to highlight similarities, differences, and broader implications.

In addition, the authors are encouraged to engage with more recent literature to strengthen the discussion. For example, recently published studies (e.g., Bin Fan et al., 2025) could be used to support, contrast, or expand upon the current findings and demonstrate the study’s importance.

Conclusion

The Conclusion section is excessively long, and several parts contain interpretative content that would be more appropriately placed in the Discussion section. The authors are encouraged to streamline the conclusion by focusing on the key findings and main take home messages, while moving detailed explanations and comparative discussion to the Discussion section.

We look forward to receiving your revised manuscript.

Kind regards,

Noé Aguilar-Rivera

Academic Editor

PLOS One

Journal Requirements:

Reviewers' comments:

Reviewer's Responses to Questions

**Comments to the Author**

1. If the authors have adequately addressed your comments raised in a previous round of review and you feel that this manuscript is now acceptable for publication, you may indicate that here to bypass the “Comments to the Author” section, enter your conflict of interest statement in the “Confidential to Editor” section, and submit your "Accept" recommendation.

Reviewer #2: (No Response)

2. Is the manuscript technically sound, and do the data support the conclusions?

Reviewer #2: Yes

3. Has the statistical analysis been performed appropriately and rigorously? 

Reviewer #2: No

4. Have the authors made all data underlying the findings in their manuscript fully available?

The PLOS Data policy requires authors to make all data underlying the findings described in their manuscript fully available without restriction, with rare exception (please refer to the Data Availability Statement in the manuscript PDF file). The data should be provided as part of the manuscript or its supporting information, or deposited to a public repository. For example, in addition to summary statistics, the data points behind means, medians and variance measures should be available. If there are restrictions on publicly sharing data—e.g. participant privacy or use of data from a third party—those must be specified.requires authors to make all data underlying the findings described in their manuscript fully available without restriction, with rare exception (please refer to the Data Availability Statement in the manuscript PDF file). The data should be provided as part of the manuscript or its supporting information, or deposited to a public repository. For example, in addition to summary statistics, the data points behind means, medians and variance measures should be available. If there are restrictions on publicly sharing data—e.g. participant privacy or use of data from a third party—those must be specified.requires authors to make all data underlying the findings described in their manuscript fully available without restriction, with rare exception (please refer to the Data Availability Statement in the manuscript PDF file). The data should be provided as part of the manuscript or its supporting information, or deposited to a public repository. For example, in addition to summary statistics, the data points behind means, medians and variance measures should be available. If there are restrictions on publicly sharing data—e.g. participant privacy or use of data from a third party—those must be specified.requires authors to make all data underlying the findings described in their manuscript fully available without restriction, with rare exception (please refer to the Data Availability Statement in the manuscript PDF file). The data should be provided as part of the manuscript or its supporting information, or deposited to a public repository. For example, in addition to summary statistics, the data points behind means, medians and variance measures should be available. If there are restrictions on publicly sharing data—e.g. participant privacy or use of data from a third party—those must be specified.

Reviewer #2: Yes

5. Is the manuscript presented in an intelligible fashion and written in standard English?

Reviewer #2: No

6. Review Comments to the Author

Reviewer #2: Introduction

In the section beginning with “Citrus, indigenous to China … until the competitiveness of the industry”, no reference is provided to support the statements made. Appropriate citations should be added to substantiate this information.

Additionally, the authors report data up to 2023; however, given that we are now in 2025, it would strengthen the manuscript to include the most recent available data, along with corresponding references, to provide an updated perspective.

Regarding the citation of Chen Xinjian et al., the reference number for this study should be clearly indicated in the text to ensure consistency with the reference list.

It is recommended that the authors avoid using the phrase “this paper” and instead use “this study” or “this research”.

Material and methods

It would improve the clarity and logical flow of this section if the authors first describe the data sources, followed by the model construction used in the study.

Results

After the Materials and Methods section, the subheading Results is missing and should be added.

For Table 1, How the standard deviation of Indirect material and service input per mu (yuan) (279.24) exceeds the corresponding mean value (267.02)? The authors should explain this variability or verify the accuracy of the reported data.

In addition, no statistical analyses are reported for Table 1, Table 3, Figure 2, and Figure 3. Interpretation or discussion of differences and significance should only be presented if appropriate statistical tests were conducted. The authors are encouraged to include the relevant statistical analyses.

Discussion

The Discussion section needs to be improved. The authors should provide a deeper interpretation of their results, discussing it with previous studies. In particular, the findings could be better contextualized by comparing them with similar studies conducted in other countries, such as Brazil, to highlight similarities, differences, and broader implications.

In addition, the authors are encouraged to engage with more recent literature to strengthen the discussion. For example, recently published studies (e.g., Bin Fan et al., 2025) could be used to support, contrast, or expand upon the current findings and demonstrate the study’s importance.

Conclusion

The Conclusion section is excessively long, and several parts contain interpretative content that would be more appropriately placed in the Discussion section. The authors are encouraged to streamline the conclusion by focusing on the key findings and main take home messages, while moving detailed explanations and comparative discussion to the Discussion section.

7. PLOS authors have the option to publish the peer review history of their article (what does this mean?). If published, this will include your full peer review and any attached files.). If published, this will include your full peer review and any attached files.). If published, this will include your full peer review and any attached files.). If published, this will include your full peer review and any attached files.

...

Reviewer #2: **Yes:** Saoussen Ben AbdallahSaoussen Ben AbdallahSaoussen Ben AbdallahSaoussen Ben Abdallah

---

## [Author Response · Author response to Decision Letter 4]

7 Feb 2026

Reviewers' comments:

Reviewer #2: Introduction

In the section beginning with “Citrus, indigenous to China … until the competitiveness of the industry”, no reference is provided to support the statements made. Appropriate citations should be added to substantiate this information.

Additionally, the authors report data up to 2023; however, given that we are now in 2025, it would strengthen the manuscript to include the most recent available data, along with corresponding references, to provide an updated perspective.

Appropriate citations have been added as requested. All data throughout the text has been updated to 2024 (the 2025 yearbook contains 2024 data), including cultivation data for the seven sample provinces and municipalities discussed later. Re-calculations and analyses have been performed based on the latest data.

Regarding the citation of Chen Xinjian et al., the reference number for this study should be clearly indicated in the text to ensure consistency with the reference list.

The revisions have been made as requested.

It is recommended that the authors avoid using the phrase “this paper” and instead use “this study” or “this research”.

The revisions have been made as requested.

Material and methods

It would improve the clarity and logical flow of this section if the authors first describe the data sources, followed by the model construction used in the study.

Given the specific metrics involved in the model, the subsequent data has been organized according to these metrics to facilitate better understanding and correspondence.

Results

After the Materials and Methods section, the subheading Results is missing and should be added.

The results section has been added; see the main text for details.

For Table 1, How the standard deviation of Indirect material and service input per mu (yuan) (279.24) exceeds the corresponding mean value (267.02)? The authors should explain this variability or verify the accuracy of the reported data.

A standard deviation greater than the mean indicates a high degree of data dispersion, the presence of outliers, and a skewed distribution. This has been appropriately explained in the relevant section.

In addition, no statistical analyses are reported for Table 1, Table 3, Figure 2, and Figure 3. Interpretation or discussion of differences and significance should only be presented if appropriate statistical tests were conducted. The authors are encouraged to include the relevant statistical analyses.

The relevant sections have been revised accordingly.

Discussion

The Discussion section needs to be improved. The authors should provide a deeper interpretation of their results, discussing it with previous studies. In particular, the findings could be better contextualized by comparing them with similar studies conducted in other countries, such as Brazil, to highlight similarities, differences, and broader implications.

In addition, the authors are encouraged to engage with more recent literature to strengthen the discussion. For example, recently published studies (e.g., Bin Fan et al., 2025) could be used to support, contrast, or expand upon the current findings and demonstrate the study’s importance.

Appropriate improvements have been made as required.

Conclusion

The Conclusion section is excessively long, and several parts contain interpretative content that would be more appropriately placed in the Discussion section. The authors are encouraged to streamline the conclusion by focusing on the key findings and main take home messages, while moving detailed explanations and comparative discussion to the Discussion section.

Appropriate edits have been made to the content of the conclusion section.

---

## [Decision Letter · Decision Letter 4]

12 Mar 2026

Evaluating the Technical Efficiency and Influencing Factors of Citrus Fruits Planting in China

PONE-D-25-04501R4

Dear Dr. yuan wu

We’re pleased to inform you that your manuscript has been judged scientifically suitable for publication and will be formally accepted for publication once it meets all outstanding technical requirements.

Kind regards,

Noé Aguilar-Rivera

Academic Editor

PLOS One

Additional Editor Comments (optional):

Reviewers' comments:

Reviewer's Responses to Questions

**Comments to the Author**

1. If the authors have adequately addressed your comments raised in a previous round of review and you feel that this manuscript is now acceptable for publication, you may indicate that here to bypass the “Comments to the Author” section, enter your conflict of interest statement in the “Confidential to Editor” section, and submit your "Accept" recommendation.

Reviewer #2: (No Response)

Reviewer #4: All comments have been addressed

2. Is the manuscript technically sound, and do the data support the conclusions?

Reviewer #2: No

Reviewer #4: Yes

3. Has the statistical analysis been performed appropriately and rigorously? 

Reviewer #2: I Don't Know

Reviewer #4: Yes

4. Have the authors made all data underlying the findings in their manuscript fully available?

The PLOS Data policy requires authors to make all data underlying the findings described in their manuscript fully available without restriction, with rare exception (please refer to the Data Availability Statement in the manuscript PDF file). The data should be provided as part of the manuscript or its supporting information, or deposited to a public repository. For example, in addition to summary statistics, the data points behind means, medians and variance measures should be available. If there are restrictions on publicly sharing data—e.g. participant privacy or use of data from a third party—those must be specified.requires authors to make all data underlying the findings described in their manuscript fully available without restriction, with rare exception (please refer to the Data Availability Statement in the manuscript PDF file). The data should be provided as part of the manuscript or its supporting information, or deposited to a public repository. For example, in addition to summary statistics, the data points behind means, medians and variance measures should be available. If there are restrictions on publicly sharing data—e.g. participant privacy or use of data from a third party—those must be specified.requires authors to make all data underlying the findings described in their manuscript fully available without restriction, with rare exception (please refer to the Data Availability Statement in the manuscript PDF file). The data should be provided as part of the manuscript or its supporting information, or deposited to a public repository. For example, in addition to summary statistics, the data points behind means, medians and variance measures should be available. If there are restrictions on publicly sharing data—e.g. participant privacy or use of data from a third party—those must be specified.requires authors to make all data underlying the findings described in their manuscript fully available without restriction, with rare exception (please refer to the Data Availability Statement in the manuscript PDF file). The data should be provided as part of the manuscript or its supporting information, or deposited to a public repository. For example, in addition to summary statistics, the data points behind means, medians and variance measures should be available. If there are restrictions on publicly sharing data—e.g. participant privacy or use of data from a third party—those must be specified.

Reviewer #2: Yes

Reviewer #4: Yes

5. Is the manuscript presented in an intelligible fashion and written in standard English?

Reviewer #2: No

Reviewer #4: Yes

6. Review Comments to the Author

Reviewer #2: The manuscript addresses an important topic: the technical efficiency of citrus planting in China using stochastic frontier analysis. The manuscript lacks a clear organizational structure, making it difficult to follow the progression from objectives to methods, results, and discussion. There is no coherent overarching message, and it is unclear how the findings can meaningfully inform strategies to improve citrus production. The manuscript suffers from fundamental weaknesses in data robustness, and results reporting that undermine its suitability for publication in its current form.

Reviewer #4: (No Response)

7. PLOS authors have the option to publish the peer review history of their article (what does this mean?). If published, this will include your full peer review and any attached files.). If published, this will include your full peer review and any attached files.). If published, this will include your full peer review and any attached files.). If published, this will include your full peer review and any attached files.

...

Reviewer #2: No

Reviewer #4: **Yes:** Christian Michel-CuelloChristian Michel-CuelloChristian Michel-CuelloChristian Michel-Cuello

---

## [Editor Report · Acceptance letter]

PONE-D-25-04501R4

PLOS One

Dear Dr. wu,

I'm pleased to inform you that your manuscript has been deemed suitable for publication in PLOS One. Congratulations! Your manuscript is now being handed over to our production team.

Kind regards,

on behalf of

Dr. Noé Aguilar-Rivera

Academic Editor

PLOS One